# Electrical and Gas Sensor Properties of Nb(V) Doped Nanocrystalline β-Ga_2_O_3_

**DOI:** 10.3390/ma15248916

**Published:** 2022-12-13

**Authors:** Matvei Andreev, Maxim Topchiy, Andrey Asachenko, Artemii Beltiukov, Vladimir Amelichev, Alina Sagitova, Sergey Maksimov, Andrei Smirnov, Marina Rumyantseva, Valeriy Krivetskiy

**Affiliations:** 1Department of Chemistry, Lomonosov Moscow State University, Leninskie Gory 1/3, 119234 Moscow, Russia; 2A.V. Topchiev Institute of Petrochemical Synthesis, Russian Academy of Sciences, Leninsky Prospect 29, 119991 Moscow, Russia; 3Udmurt Federal Research Center of the Ural Branch of the Russian Academy of Sciences, Tatyana Baramzina St. 34, 426067 Izhevsk, Russia; 4Scientific-Manufacturing Complex «Technological Centre», Shokina Square, House 1, Bld. 7 Off. 7237, 124498 Zelenograd, Moscow, Russia

**Keywords:** Ga_2_O_3_, metal oxide, semiconductor, gas sensor, ultrafine state, doping, niobium, defect

## Abstract

A flame spray pyrolysis (FSP) technique was applied to obtain pure and Nb(V)-doped nanocrystalline β-Ga_2_O_3_, which were further studied as gas sensor materials. The obtained samples were characterized with XRD, XPS, TEM, Raman spectroscopy and BET method. Formation of GaNbO_4_ phase is observed at high annealing temperatures. Transition of Ga(III) into Ga(I) state during Nb(V) doping prevents donor charge carriers generation and hinders considerable improvement of electrical and gas sensor properties of β-Ga_2_O_3_. Superior gas sensor performance of obtained ultrafine materials at lower operating temperatures compared to previously reported thin film Ga_2_O_3_ materials is shown.

## 1. Introduction

Metal oxide semiconductor gas sensors find their applications in many important fields: industrial safety [1], ecological monitoring [2], indoor air quality assessment [3], non-invasive medical diagnostics [4] and others [5,6]. Significant drawback of this type of sensors—low selectivity of response (high cross-sensitivity)—can be overcome by utilization of machine learning algorithms, which have been proposed on either single sensor, or multi-sensor systems [7,8,9]. Additionally, machine learning techniques allow to partially compensate response instability and drift effects [10]. However, the stability of intrinsic electrical and chemical characteristics of gas sensor material is still very important, otherwise more or less frequent recalibration MOX gas sensor operating devices is required [11]. Insufficient sensor stability is caused by diffusion-related inter-grain mass transfer, phase transitions, surface poisoning and variable environmental conditions, such as moisture level in working media [12].

As a semiconductor material Ga_2_O_3_ materials were first proposed by Fleischer et al. to detect oxygen concentration in atmosphere [13,14]. However, gallium oxide β-Ga_2_O_3_ shows n-type semiconductor properties only at high temperatures due to band gap width about 4.9 eV, while exact value varies depending on material physical state [15]. As a result, thin gas sensing films of Ga_2_O_3_ usually show their optimum performance towards either oxidizing or reducing gases, among which are O_2_, NO, O_3_, CO, CH_4_ and other hydrocarbons, ammonia and volatile organic compounds, only at high temperatures over 450 °C [16]. Efforts to decrease operating temperatures of gallia-based gas sensors are directed at composite sensitive materials formation [17,18,19], decoration with noble metals, possessing catalytic properties [20,21] or exploitation of size effect of nanocrystalline high surface area Ga_2_O_3_-based materials [22,23,24]. The latter strategy may be considered as the most beneficial as the introduction of the second phase, either metal oxide or noble metal, in the materials structure can compromise the desired long-term stability of material and reliability of manufacturing process [25,26]. Doping of Ga_2_O_3_ with n-type donor dopants—Sn, Ti and Si—has been reported to improve its electrical and gas sensing properties towards both oxidating and reducing gases due to activation of oxygen surface chemisorption [27,28,29,30]. Other n-type dopants have been reported in experimental and theoretical studies to drastically improve electrical properties of Ga_2_O_3_, valuable in semiconductor industry [31,32,33]. Among them Nb(V) has been proposed as a perspective and effective n-type dopant, being a shallow donor with small ionization energy and close ionic radii to Ga(III)in both octahedral and tetrahedral positions of monoclinic β-Ga_2_O_3_ lattice [34]. Further experimental studies have supported this concept [31,35,36].

Thus, the Nb-doped β-Ga_2_O_3_ may be suggested as a very promising highly stable material for semiconductor gas sensors; however, to the best of the authors’ knowledge, the ability to the date neither thin film materials nor various ultrafine or nano-dimensional moieties has not been studied in this regard. The objective of the present study is to investigate the structural, electrical and gas sensor properties of β-Ga_2_O_3_ in ultrafine state and describe the effects of Nb(V) doping on the properties of β-Ga_2_O_3_ nano-crystals.

## 2. Materials and Methods

### 2.1. Materials Synthesis

Ga_2_O_3_/Nb materials were obtained via flame spray pyrolysis technique, thoroughly described in previous paper [37]. This synthetic approach has been chosen as it allows to obtain metal oxide materials in ultrafine state with homogeneous distribution of additional components—whether doping or surface modifying ones [38]. Toluene (99.9%) was used as a fuel. Following metalorganic precursors were used: gallium acetylacetonate and niobium 2-ethylhexanoate (V) (GELEST INC, 95%). Gallium acetylacetonate was prepared by mixing potassium acetylacetonate with gallium nitrate Ga(NO_3_)_3_·8H_2_O (Rare Metals Plant, Russia, 99.4%) in stochiometric ratio followed by purification with dichloromethane solution.

An amount of 0.2 M solutions of precursors in toluene were prepared with intended concentration of Nb (V) cation: 0, 1, 2 and 4 mol.% of common cation quantity. The liquid mixture was supplied to spray nozzle at 3 mL/min rate. Mixture was sprayed with oxygen (99.9%) flow of 1.5 L/min at pressure drop of 3 atm. To ignite the aerosol a circular methane/oxygen flame was used. The materials were collected on glass fiber filters (GE Whatman, Sigma-Aldrich, St. Louis, MO, USA), located 80 cm above nozzle with the aid of vacuum pump ISP 250 C (Anest Iwata, Yokohama, Japan). Powders were collected from filter manually and heated in tubular furnace at specified temperatures (500–1000 °C) for 24 h in dry clean air flow (300 mL/min). Annealed samples were characterized with the set of methods and used gas for sensor fabrication. Obtained materials (Table 1) were labeled as Ga_2_O_3_/Nb_x_y, where “x” and “y” represent given Nb(V) content and annealing temperature (°C) respectively (absence of “y” corresponds to as prepared sample).

### 2.2. Materials Characterization

Phase composition was studied by X-ray diffraction (XRD) using DRON-4M (Burevestnik, St. Petersburg, Russia) diffractometer with Cu K_α_ radiation (λ = 1.5406 Å) in 2θ = 5°–80° with 0.05° 2θ step. Particle sizes were calculated from Scherrer formula in spherical particles assumption. Specific surface area was measured by low temperature N_2_ adsorption (BET model) using Micromeritics Chemisorb 2750 (Micromeritics, Norcross, GA, USA). The morphology of the samples was characterized by using a JEOL JEM-2100 F/Cs/GIF/EDS transmission electron microscope (JEOL, Tokyo, Japan). Raman spectra were collected on i-Raman Plus spectrometer (BW Tek, Plainsboro Township, NJ, USA). Electronic structure was investigated on Perkin-Elmer Lambda-950 spectrometer in diffusion reflectance mode. Finally, samples composition was determined by X-Ray photoelectron spectroscopy (XPS) K-Alpha (Thermo Fisher Scientific, Waltham, MA, USA) and SPECS (SPECS, Berlin, Germany). Charge neutralization was applied, providing the C 1 s peak position of 285.0 eV.

### 2.3. Surface Reactivity Assessment

Oxidation reaction was conducted in the catalytic mode with the use of USGA-1 (Unisit, Russia) setup in order to measure surface activity of obtained materials. Carbon monoxide oxidation with oxygen was used as a model process. Samples were preliminarily heated in atmosphere of oxygen (99.9%) during 30 min at temperature of 500 °C, then cooled down to room temperature. Then, a gas mixture containing 79.6% N_2_, 15.9% O_2_ and 4.5% CO, respectively, was passed through reaction tube with samples heated with constant rate of 10 °C per minute till temperature of 900 °C. Mass-spectra during experiment were registered by means of MS7-100 mass-spectrometer (IAP RAN, St. Petersburg, Russia). Temperature of half conversion t_50 was used as a quantitative measure of surface reactivity and it was defined as:p_(t_50)_ = (p_max_ − p_min_)/2,(1)
where p_max_ and p_min_ are values, which correspond to maximum and minimum pressures of CO_2_ during experiment respectively.

### 2.4. Gas Sensor Measurements

Gas sensors were prepared by deposition of powder dispersion in binding substance α-terpineol on a heated substrate—aluminum polycrystalline wafer 2 × 2 × 0.15 mm with platinum heater and sensor electrodes as described before [37]. The binding substance was removed by the heating of the substrate at 500 °C for 24 h in the laboratory ambient air.

The experiment was conducted on laboratory made setup with ability to both control sensor temperature and measure sensor resistance [10]. Sensors were placed in a gastight flow-through PTFE chamber. Gas mixtures were prepared by diluting certified reference gas mixtures with pure dry air from a clean air generator, GCHV-2.0 (Himelectronica, Russia). To measure sensor response, gas mixture and pure air were consequently flown through sensor chamber and sensor response was calculated according to equation:S = (R_0_ − R)/R_0_,(2)
where R—is a sensing element resistance in the target gas presence, R_0_—sensor resistance in clean air. To investigate sensor properties several gas mixtures with fixed target gas concentrations were used: H_2_—20 ppm, CH_3_OH—20 ppm, acetone—20 ppm, and CH_4_—10,000 ppm. Three consequent experiments with one analyte concentration were conducted to get statistically significant results.

## 3. Results and Discussion

### 3.1. Materials Structure and Chemical Composition

XRD patterns (Figure 1) of synthesized materials show that β-Ga_2_O_3_ (ICDD PDF2 database card number 43-1012) is the primary phase in all samples. Admixture of GaNbO_4_ phase (ICDD PDF2 database card number 72-1666) is detected in samples annealed at temperatures over 900 °C with more than 2 mol.% Nb content. Solid solution boundaries were estimated by additional annealing at 800 °C, 850 °C and 1000 °C, respectively. For samples with less than 2 mol.% Nb content, no formation of an X-ray detectable GaNbO_4_ phase was shown up to 1000 °C.

The value of grain size derived from Scherrer formula decreases with growth of Nb content (Table 1). The obtained grain sizes correlate with particle dimensions measured on the basis of TEM images (Figure 2). TEM micrographs also indicate some agglomeration of nanometer size particulate matter. Elevation of annealing temperature leads to increase in particle size.

It can be generalized that the biggest specific surface area corresponds to samples with 1 mol.% Nb(V) content at every applied annealing temperature (Table 1). It should be noted that at 1000 °C annealing temperature, specific surface area value sharply decreases. This effect is probably connected with formation of GaNbO_4_ phase on particle surface: lower melting temperature of GaNbO_4_ (below 1700 K) [39] in comparison to Ga_2_O_3_ (above 2000 K) [40] leads to higher ion mobility and enhancement of diffusion processes.

Formation of GaNbO_4_ phase is supported by Raman spectroscopy data. Raman spectra (Figure 3) are given only for samples annealed at temperature over 900 °C because of intensive luminescence, due to high defect concentration in the crystal structure of materials calcined at lower temperatures.

Data obtained contains informative peaks in Raman shift range from 200 to 1200 cm^−1^. Peaks highlighted with red zones corresponds to valence and deformation oscillation modes in GaO_4_ and GaO_6_ polyhedrons. Peaks at 262 cm^−1^, 431 cm^−1^, 913 cm^−1^ should be attributed to O–Ga–O, O–Nb–O oscillation frequencies in GaNbO_4_ structure [39]. No presence of Nb_2_O_5_ was shown using Raman spectroscopy [41]. It is known that GaO_6_ octahedra distortion results in 913 cm^−1^ peak shift to lower frequency area [39]. Thus, it can be assumed that 830 cm^−1^ peak originated from Nb_Ga_ replacement defect formation in strongly distorted oxygen coordination.

The chemical composition data obtained from XPS measurements are given in Table 2. As prepared samples and materials annealed at relatively low temperature of 500 °C demonstrate Nb content, which is corresponding to initial Nb loading. Some deviations may be caused by accuracy restrictions of XPS-quantitative analysis Nb and inaccuracy of commercially available precursor description as it was used without any pretreatment or additional analysis of precise Nb content. Table 2 follows the increase in the annealing temperature from 500 °C to 900 °C, which leads to the Nb content growth in the same material—Ga_2_O_3_/Nb-4. It may be speculated upon these results, that higher annealing temperature causes migration of Nb(V) cations to crystal surface with formation of GaNbO_4_ phase. This conclusion can be done keeping in mind that according to XRD analysis for samples annealed at temperatures below 800 °C particle radius is estimated to be less than 3 nm, therefore X-Ray photoelectron spectra for these materials describe composition of nearly whole particle for these samples. For the samples, annealed at higher temperatures, XPS data collected from surface layer is about 2 nm thick.

It is also possible to investigate fine chemical state of Ga cations using XPS data (Figure 4). Though the main Ga3d component (E_b_ = 20.4 eV) peak is overlapped with the O2s peak (Eb = 23.5 eV), a third component corresponding to the Ga(I) cation state is observable for samples containing niobium [42]. The ratio of XPS peak areas of Ga(I) to Ga(III) is close to one to eight. Hence formation of solid solution occurs with charge compensation due to transformation from Ga(III) to Ga(I) and without the rise of additional free charge carriers. Usually, the O1s XPS peak is used to estimate quantity of lattice and surface O-ions [43]. In this study, the quality of O1s peaks were insufficient to clearly distinguish two different forms of O_2_-contributions.

### 3.2. Optical Band Gap

Spectra of optical adsorption (Figure 5) of obtained materials consist of one or two linear regions in coordinates (F(R)·E)^2^ to E, usually used for direct bandgap semiconductors.

High-energy linear region corresponds to bandgap width of Ga_2_O_3_ phase, while low-energy linear region should be attributed to inner bandgap transition. Notably, a low energy linear region is not observed in the case of materials, for which the GaNbO_4_ phase formation with E_g_ equal to 3.41 eV [44] is confirmed either by XRD or Raman spectroscopy. The band gap width decreases with the growth of Nb content due to formation of the Nb_Ga_ substitution defects, which create either shallow donor levels beneath the conduction band or additional oxygen vacancies levels. The modern theoretical studies indicate that the former case is more likely to be the cause of band gap narrowing [34]. Considering the gas sensing properties shallow donor levels, associated with the Nb_Ga_ defects, should positively affect the free charge carrier concentration, which might be beneficial for oxygen chemisorption on the materials surface and, as a result, for the gas sensor response itself [27,28,29,30]. The increase in the annealing temperature causes the growth of the band gap to increase the levels, corresponding to the bulk β-Ga_2_O_3_ crystal.

### 3.3. Surface Reactivity

Increase in Nb content in the obtained materials is accompanied by the growth of the temperature of CO half-conversion (Figure 6).

The observed decrease in the materials reactivity can be connected with the decrease of the CO chemisorption active sites concentration on the surface of β-Ga_2_O_3_ grains, which are mainly Ga^3+^ cations with unsaturated coordination environment [45]. The access of gas molecules to these centers is blocked by newly formed Nb-rich GaNbO_4_ phase particles. There is scarce evidence of this phase’s reactivity in the heterogeneous chemical interaction with gas molecules; however, its poor activity in photochemical processes is reported [46].

### 3.4. Gas Sensor Properties

The typical plot of an electrical resistance variation during a gas sensor response measurement experiment of gas sensor material is given in Figure 7.

All obtained materials become more conductive as a result of interaction with reducing gas molecules. It indicates that the sensor response is formed as a result of free electrons release during a process of chemical oxidation on the surface of metal oxide grains:R + [O^2−^] → RO + 2e^−^(3)
where R is a reducing gas molecule, [O^2−^] is a mobile oxygen ion in the lattice of metal oxide nanocrystal at the surface, RO is a molecular oxidation product, e^−^ is a free electron. The Equation (3) describes the oxidation by lattice oxygen as it was reported previously for heterogeneous oxidation for gallia-based materials [47].

The gas sensing properties of the obtained series of β-Ga_2_O_3_-based materials are generalized in Figure 8.

Low concentrations of the reducing gases cause the maximum of the sensor response, in the case of the materials annealed at 900 °C with the decrease in response after further annealing at higher temperature of 1000 °C. This response maximum is based on the counteraction of two factors connected with materials structure and morphology. The first one is associated with high concentration of defects in the crystal structure of materials grains, which lead to the formation of numerous levels inside the band gap acting as an electron trap and diminishing electrical and, as a result, sensor properties of the materials [48]. These defects are proven by the intense luminescence, observed during Raman spectroscopy investigation and mentioned above. The increase of annealing temperature according to XRD data allows to improve grains crystallinity, which is accompanied by gas sensory properties enhancement. On the other hand, the increase in the annealing temperature leads to the shrinkage of the effective surface area of the materials (Table 1), negatively affecting the intensity of the “solid-gas” chemical interaction. The second factor becomes dominating, when high concentrations of reducing gases are concerned. It is illustrated by the temperature dependence of sensor response towards 1000 ppm of methane, calcined below 800 °C, which shows the maximum for materials.

The influence of Nb content on the sensor response is less obvious; however, it can be generalized that those materials with 1% mol of Nb possess the highest sensor response towards reducing gases. This phenomenon can be connected to the maximum of effective surface area in the case of these materials in each series of materials obtained at the same annealing temperature. Table 3 compares the observed resistive type sensor response to most relevant reports in literature including Ga_2_O_3_ nanowires (NWs), thin films (TF) and thick films (ThF), including composites where gallia is a main component.

Table 3 leads to the conclusion that ultrafine β-Ga_2_O_3_ possesses a lower temperature of maximum sensor response compared to other forms of this material; however, in some cases, this comparison is hard to effectuate due to very different sensor response measurement conditions. Another notion is that β-Ga_2_O_3_, either described in this study or in reports of other scholars, is better suited for VOCs detection rather than simple gases, such as hydrogen or methane. The reason is that the mechanism of sensor response relies on the mobility of lattice oxygen ions at the surface of β-Ga_2_O_3_ grains, rather than chemisorbed oxygen species [16]. This mechanism requires stronger adsorption of reducing gas molecule, which is better fulfilled in case of bigger organic molecules with polar functional groups.

As the Nb-doping performed in this study does not allow for significant altering of electrical and chemical properties of the β-Ga_2_O_3_ material due to Ga(III) to Ga(I) transition, it does not affect the sensing mechanism as well. The pure and Nb-doped materials have similar dependence of sensor response not only on the working temperature, but on the gas concentration and air humidity as well (Figure 9, Figure 10 and Figure 11).

The Nb-doped material performs slightly better in the terms of absolute sensor response; however, it has similar sensitivity—an increase in the sensor response as a result of detected gas concentration increment. The observed dependence of gas sensor response on the acetone concentration slightly deviates from the linearity in the studied concentration range in accordance with power law, established for metal oxide gas sensors, operating in a DC mode [56,57]. Additionally, it suffers from the negative humidity influence the same way the pure β-Ga_2_O_3_ does. Once humidity is introduced to the analyzed media, even in relatively small amounts,—only 2%—the sensor response drops down two times. However, a further increase in the humidity does not lead to such dramatic changes in the response value, while the working temperature of maximum sensor response slightly shifts to higher values. Keeping in mind the data on the catalytic performance of the materials reported in Section 3.3, one should conclude the higher sensor response of Ga_2_O_3_/Nb-1-900 sample to be the result of grain size and effective surface area effect.

However, considering the long-term stability of obtained materials, it should be noted that they indeed show no degradation of response during continuous exploitation (Figure 12 and Figure 13). Instead, the opposite trend is observed—an improvement in response after many days of operation.

This effect should be explained as the improvement of the gas sensing thick film layer consistency and intergrain contact number increase over the time of operation. As the XRD and TEM data has shown, the 300–500 °C temperature range is not enough to cause Ga_2_O_3_ grain growth, which may diminish the size effect and sensor response as a result. However, some minute surface diffusion processes were present during repetitious surface partial reduction and further reoxidation in the course of gas sensor response measurement towards reducing gases and VOCs. These processes led to formation of new “necks” between Ga_2_O_3_ grains, the electron transport through which is controlled by the presence of reducing gases and, thus, contributed to the sensor response growth over time [58,59]. Additional long term gas sensor measurements are needed to investigate whether these slow diffusion related processes may lead to grain and intergrain contact growth, resulting in gas sensor response degradation or the sensing porous film stabilization at some point, which is crucial for the practical application.

## 4. Conclusions

The use of flame spray pyrolysis process allows for β-Ga_2_O_3_ to be obtained in an ultrafine state with 6–12 nm grain size and the effective surface area more than 100 m^2^/g. Doping by Nb(V) during the synthetic procedure leads to both Nb incorporation in the lattice of β-Ga_2_O_3_ grains and formation of separate GaNbO_4_ phase if the temperature of the post-synthetic annealing goes above 800 °C. Doping of β-Ga_2_O_3_ by Nb(V) is accompanied by the charge compensation of the cationic sublattice by means of Ga(III) to Ga (I) transition. The combination of these phenomena does not allow to significantly improve electrical and gas sensor properties of ultrafine β-Ga_2_O_3_ by doping with electron-donor cation Nb(V). The observed enhancement of the β-Ga_2_O_3_-based gas sensor performance can be connected to the increase in materials surface area due to the hampering of diffusion related grains growth and agglomeration at elevated working temperatures. The highest sensor response to low concentrations of reducing gases is observed in the case of Ga_2_O_3_ doped with 1% mol Nb(V) and passed through 900 °C 24 h postsynthetic annealing. The obtained ultrafine nanocrystalline samples are most suited to detection of VOCs and allowed the working temperature to decrease and sensor response to increase compared to other forms of Ga_2_O_3_—thin films, nanowires or thick films. The manufactured sensing elements demonstrated improvement of gas sensor response during long term exploitation due to enhanced intergrain contacts and porous film integrity. Modification of the synthetic procedure is required in order to obtain Nb(V) doped Ga_2_O_3_ in ultrafine nanocrystalline state without Ga(III) to Ga (I) transition.

## Figures and Tables

**Figure 1 materials-15-08916-f001:**
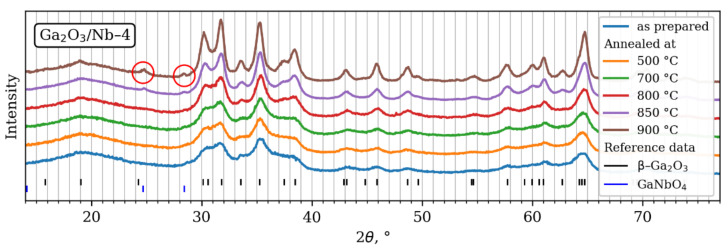
XRD patterns of synthesized materials with highest Nb content and analysis of phase composition.

**Figure 2 materials-15-08916-f002:**
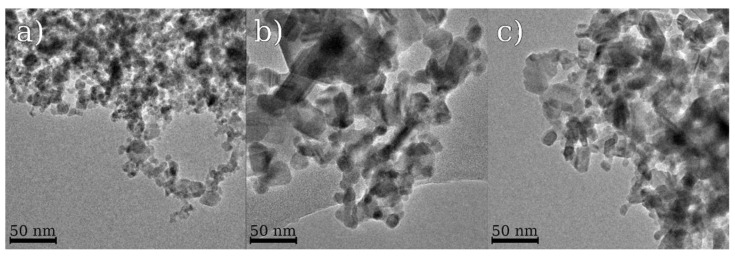
TEM images of samples: (**a**) Ga_2_O_3_/Nb-0-500; (**b**) Ga_2_O_3_/Nb-0-900; (**c**) Ga_2_O_3_/Nb-1-900.

**Figure 3 materials-15-08916-f003:**
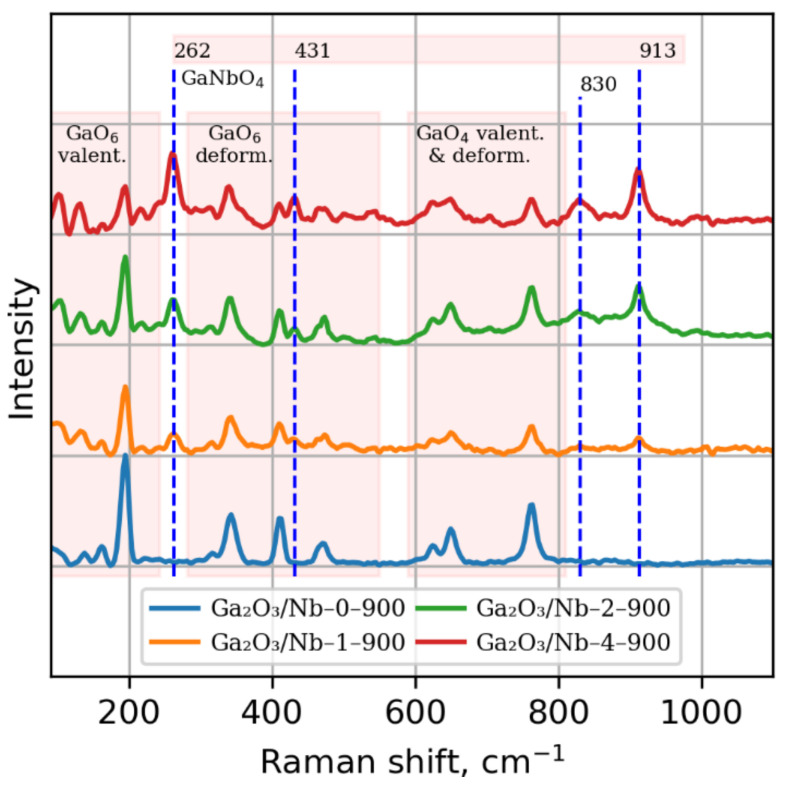
Raman spectra of Ga_2_O_3_/Nb-x-900 materials. Dash line marked peaks correspond to lattice oscillations in GaNbO_4_ phase. Remaining peaks are attributed to valence and deformation lattice oscillations in β-Ga_2_O_3_ phase.

**Figure 4 materials-15-08916-f004:**
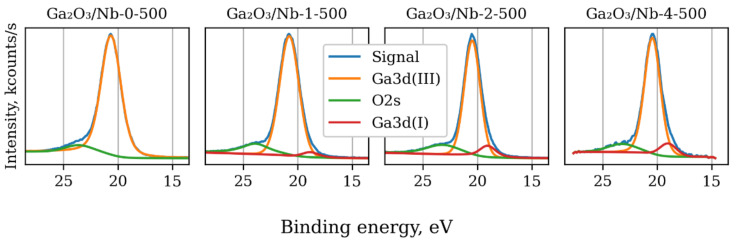
Deconvolution of Ga3d XPS spectra of materials with different Nb content and same annealing temperature.

**Figure 5 materials-15-08916-f005:**
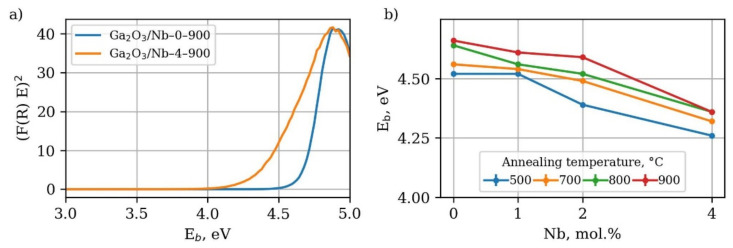
(**a**) Kubelka–Munk plot of optical absorption of synthesized materials (**b**) dependence of optical band gap width on Nb content and annealing temperature.

**Figure 6 materials-15-08916-f006:**
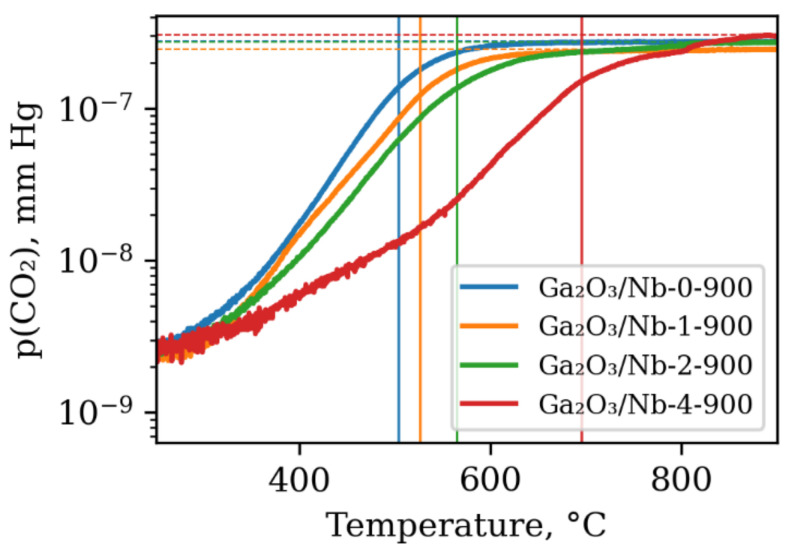
Dependency of partial CO_2_ pressure at the reaction chamber outlet on sample temperature during CO oxidation in the presence of oxygen. Vertical lines indicate temperature of CO_2_ half conversion and are 503 °C, 526 °C, 564 °C and 695 °C for pure β-Ga_2_O_3_ and 1,2,4 mol.% Nb containing materials, respectively.

**Figure 7 materials-15-08916-f007:**
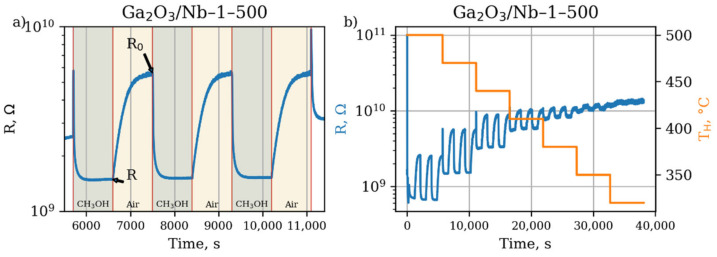
Dependence of Ga_2_O_3_/Nb-1-500 sensor material resistance on experiment time during the process of response measurement towards 20 ppm of methanol in dry air at (**a**) fixed working temperature 470 °C (**b**) different sensor working temperatures. Designations “R” and “R_0_” on the Figure 7a indicate the resistance of sensitive layer in the presence of methanol 20 ppm and in the flow of clean air, respectively, which are used to calculate sensor response according to Equation (2).

**Figure 8 materials-15-08916-f008:**
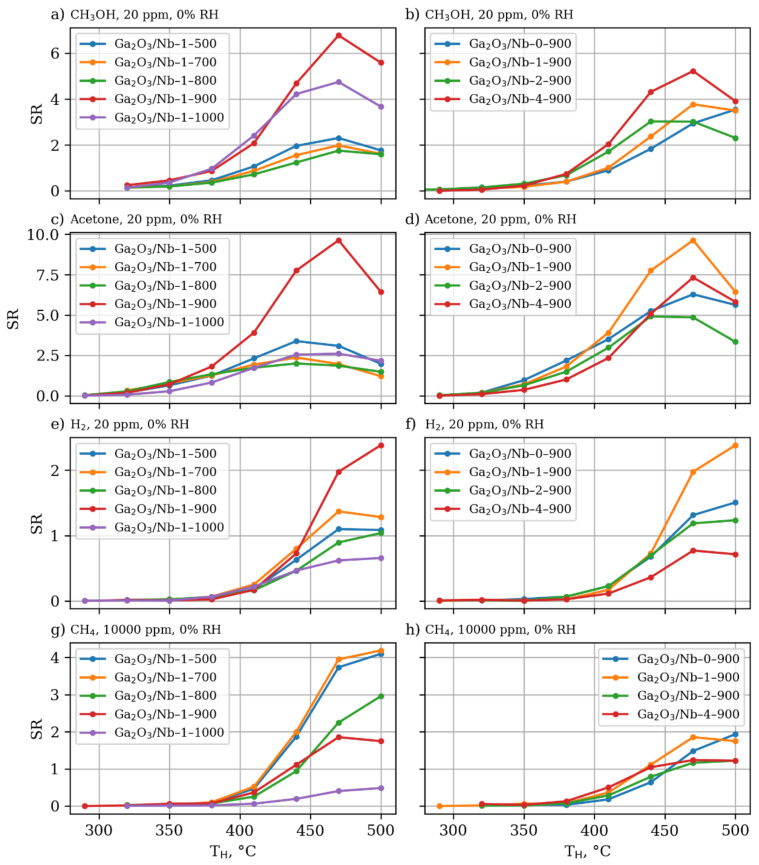
Dependence of sensor response of Ga_2_O_3_-based materials towards reducing gases and volatile organic compounds (VOCs) on temperature in relation to materials annealing temperature and Nb content—(**a**) sensor response of Ga_2_O_3_-based materials doped with 1% mol Nb and annealed at different temperatures towards 20 ppm of methanol in dry air (**b**) sensor response of Ga_2_O_3_-based materials doped with various mol content of Nb and annealed at 900 °C temperature towards 20 ppm of methanol in dry air (**c**) sensor response of Ga_2_O_3_-based materials doped with 1% mol Nb and anneald at different temperatures towards 20 ppm of methanol in dry air (**d**) sensor response of Ga_2_O_3_-based materials doped with various mol content of Nb and annealed at 900 °C temperature towards 20 ppm of acetone in dry air (**e**) sensor response of Ga_2_O_3_-based materials doped with 1% mol Nb and anneald at different temperatures towards 20 ppm of hydrogen in dry air (**f**) sensor response of Ga_2_O_3_-based materials doped with various mol content of Nb and annealed at 900 °C temperature towards 20 ppm of hydrogen in dry air (**g**) sensor response of Ga_2_O_3_-based materials doped with 1% mol Nb and anneald at different temperatures towards 10,000 ppm of methane in dry air (**h**) sensor response of Ga_2_O_3_-based materials doped with various mol content of Nb and annealed at 900 °C temperature towards 10,000 ppm of methane in dry air.

**Figure 9 materials-15-08916-f009:**
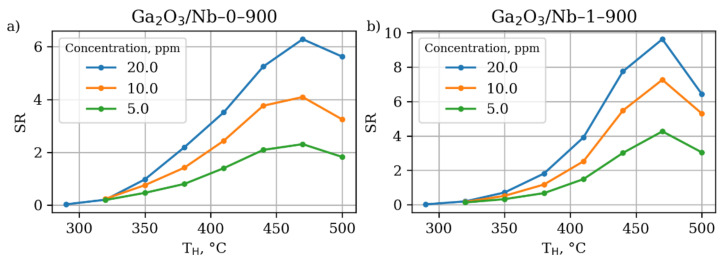
Dependence of sensor response of (**a**) Ga_2_O_3_-based materials without Nb doping annealed at 900 °C and (**b**) Ga_2_O_3_-based materials with 1% mol Nb doping annealed at 900 °C at different working temperatures towards acetone on concentration in dry air.

**Figure 10 materials-15-08916-f010:**
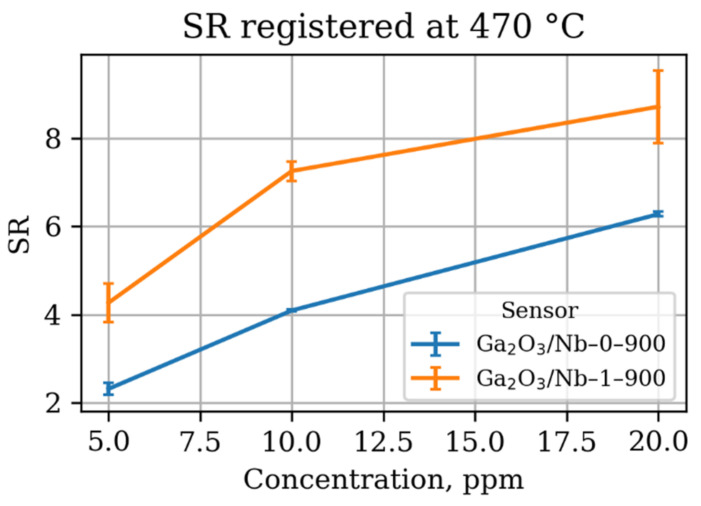
Dependence of sensor response of Ga_2_O_3_-based materials at 470 °C working temperature towards acetone on its concentration in dry air.

**Figure 11 materials-15-08916-f011:**
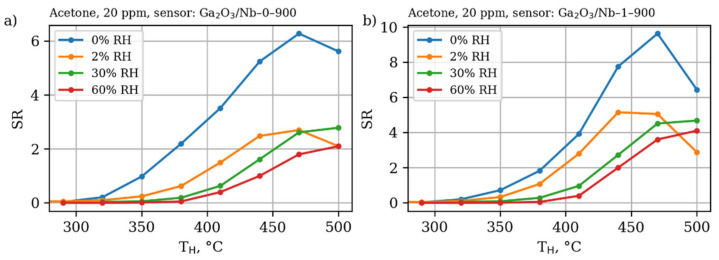
Dependence of sensor response of (**a**) pure Ga_2_O_3_-based materials annealed at 900 °C and (**b**) 1% mol Nb-doped Ga_2_O_3_-based materials annealed at 900 °C at different working temperatures towards acetone 20 ppm in dry air and in air with relative humidity (RH) 2%, 30% and 60%.

**Figure 12 materials-15-08916-f012:**
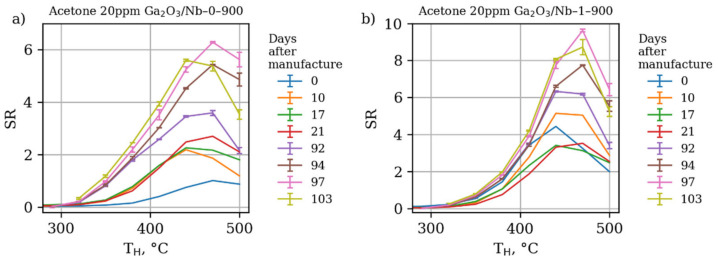
Dependence of sensor response of (**a**) pure Ga_2_O_3_-based materials annealed at 900 °C and (**b**) 1% mol Nb-doped Ga_2_O_3_-based materials annealed at 900 °C at different working temperatures towards acetone 20 ppm in dry air at different days after manufacturing and start of gas sensor response study. The sensors were busy in the 300–500 °C working temperature range almost every day since the first day of operation.

**Figure 13 materials-15-08916-f013:**
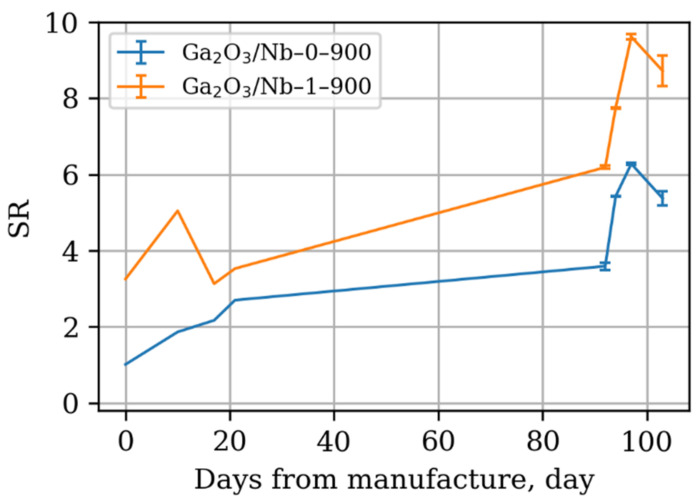
Dependence of sensor response of Ga_2_O_3_-based materials at fixed working temperature 470 °C towards acetone 20 ppm in dry air at different days after manufacturing and start of gas sensor response study.

**Table 1 materials-15-08916-t001:** Morphological parameters of the obtained materials—grain size, nm/surface area—m^2^/g.

Annealing Temperature, °C (y)	Nb Content, mol.% (x)			
	0	1	2	4
-	6/97	6/111	6/92	6/109
500	6/104	6/127	6/102	6/103
700	7/93	7/98	7/95	6/88
800	9/63	8/73	7/65	7/67
900	12/24	11/49	10/41	11/34
1000	15/11	19/1	-	-

**Table 2 materials-15-08916-t002:** Nb content in the synthesized samples according to XPS.

Sample	Nb Content, mol.% (x)
Ga_2_O_3_/Nb-0-500	0
Ga_2_O_3_/Nb-1-500	1.6
Ga_2_O_3_/Nb-2-500	2.6
Ga_2_O_3_/Nb-4	4.2
Ga_2_O_3_/Nb-4-500	3.9
Ga_2_O_3_/Nb-4-800	5.4
Ga_2_O_3_/Nb-4-900	8.7

**Table 3 materials-15-08916-t003:** Sensor response of various Ga_2_O_3_-based materials towards gases and VOCs *.

Material	Gas	T_w_, °C	S, a.u.	Reference
Ga_2_O_3_/Nb-1-900	Acetone, 20 ppm	470	9.8	This work
Ga_2_O_3_ NWs	Acetone, 100 ppm	600	40	[49]
Ga_2_O_3_ ThF	Acetone, 200 ppm	480	35	[50]
Ga_2_O_3_ TF	Acetone, 1600 ppm	530	100	[51]
Ga_2_O_3_/Nb-1-900	H_2_, 20 ppm	500	2.5	This work
Ga_2_O_3_-Cr_2_O_3_ TF	H_2_, 2500 ppm	500	60	[19]
Ga_2_O_3_ ThF	H_2_, 5000 ppm	500	1	[52]
H_2_, 20 ppm	H_2_, 20 ppm	700	2	[53] **
Ga_2_O_3_/Nb-1-500	CH_4_, 10,000 ppm	500	4	This work
Ga_2_O_3_ TF	CH_4_, 5000 ppm	840	100	[54] **
Ga_2_O_3_ ThF	CH_4_, 10,000 ppm	570	10	[55] **

* No methanol sensing data with Ga_2_O_3_ resistive sensors were found by authors in the literature. ** The most relevant studies of many publications by M. Fleischer et al. are cited considering used H_2_ and CH_4_ concentrations, measurement conditions and studies similarity.

## Data Availability

Not applicable.

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
