# Peer review of "Electrical and Gas Sensor Properties of Nb(V) Doped Nanocrystalline β-Ga2O3"

_materials, 2022, doi:10.3390/ma15248916_

Round 1

Reviewer 1 Report

The article »Electrical and gas sensor properties of Nb(V) doped nanocrystalline β-Ga2O3« by Matvei Andreev et al. is taking good advantage of Ga2O3 based material. The manuscript is well written in terms of content. Also, the corresponding explanation are full and reasonable. This reviewer consider that the paper is suitable for publication after careful modification. My views are listed blew:

1.First and foremost, from the viewpoint of the article’s structure, the main space is on the characterization and analysis of materials, while there have been many studies on Nb(V) doped β-Ga2O3. The research on gas sensor is relatively insufficient, especially regarding the analysis of gas sensor mechanism. So, authors need to explain exactly the innovation and/or Nb(V)-β-Ga2O3 based gas sensor of this work.

2. Although there are many references, most of them are outdated and cannot reflect the latest research progress. Only 3 articles published in 2022 are included in all 43 references.

3.Table 2. Nb content in the synthesized samples according to XPS. Generally speaking, doping more than 5% will be difficult, and the authors need to explain what's the point of this and how such a high doping concentration benefits the improvement of gas sensor.

4. Page 8/9 of 11, From the author's statement, it does not reflect the outstanding advantages of using Ga2O3 as the base-material. Compared to other oxides, the author needs to explain or cite papers what advantages Ga2O3 has when used in gas sensor.

Author Response

 1.First and foremost, from the viewpoint of the article’s structure, the main space is on the characterization and analysis of materials, while there have been many studies on Nb(V) doped β-Ga2O3. The research on gas sensor is relatively insufficient, especially regarding the analysis of gas sensor mechanism. So, authors need to explain exactly the innovation and/or Nb(V)-β-Ga2O3 based gas sensor of this work.

The introduction to the report has been rewritten in order to better explain the motivation to this research:

“…However, gallium oxide β-Ga2O3 shows n-type semiconductor properties only at high temperatures due to band gap width about 4.9 eV, while exact value varies depending on material physical state [15]. As a result, thin gas sensing films of Ga2O3 usually show their optimum performance towards either oxidizing or reducing gases, among which O2, NO, O3, CO, CH4 and other hydrocarbons, ammonia and volatile organic compounds only at high temperatures over 450 oC [16]. Efforts to decrease operating temperatures of gallia-based gas sensors are directed at composite sensitive materials formation [17-19], decoration with noble metals, possessing catalytic properties [20, 21], or exploitation of size effect of nanocrystalline high surface area Ga2O3-based ma-terials [22-24]. The latter strategy may be considered as the most beneficial as the in-troduction of the second phase, either metal oxide or noble metal, in the materials structure can compromise the desired long-term stability of material as well as relia-bility of manufacturing process [25, 26]. Doping of Ga2O3 with n-type donor dopants – Sn, Ti, Si - has been reported to improve its electrical and gas sensing properties to-wards both oxidating and reducing gases due to activation of oxygen surface chemi-sorption [27-30]. Other n-type dopants have been reported in experimental and theo-retical studies to drastically improve electrical properties of Ga2O3, valuable in semi-conductor industry [31-33]. Among them Nb(V) has been proposed as a perspective and effective n-type dopant, being a shallow donor with small ionization energy and close ionic radii to Ga(III)in both octahedral and tetrahedral positions of monoclinic β-Ga2O3 lattice [34]. Further experimental studies have supported this concept [31, 35, 36]….”

  1. Although there are many references, most of them are outdated and cannot reflect the latest research progress. Only 3 articles published in 2022 are included in all 43 references.

The latest relevant references were added and out timed or irrelevant ones were removed.

 3.Table 2. Nb content in the synthesized samples according to XPS. Generally speaking, doping more than 5% will be difficult, and the authors need to explain what's the point of this and how such a high doping concentration benefits the improvement of gas sensor.

The authors respectfully agree with the opinion of the reviewer, that is why 1, 2 and 4% mol of Nb was loaded in the obtained samples. Table 2 is given in in the text of manuscript with the purpose to demonstrate the growth of Nb content, quantified of XPS data, alongside with the increase of temperature of annealing. It supports the notion on the Nb-containing phase segregation on the grains surface during high temperature treatment in the case of high initial loading of Nb. The necessary explanations are added to the text:

«…The chemical composition data obtained from XPS measurements are shown given in table 2. As prepared samples and materials annealed at relatively low temperature of 500 oC demonstrate Nb content, which is corresponding to initial Nb loading. Some deviations may be caused by accuracy restrictions of XPS-quantitative analysis Nb and inaccuracy of commercially available precursor description as it was used without any pretreatment or additional analysis of precise Nb content. From the Table 2 it follows that increase in the annealing temperature from 500 up to 900 oC leads to the Nb content growth in the same material - Ga2O3/Nb-4. It may be speculated upon these results, that higher annealing temperature causes migration of Nb(V) cations to crystal surface with formation of GaNbO4 phase…»

  1. Page 8/9 of 11, From the author's statement, it does not reflect the outstanding advantages of using Ga2O3 as the base-material. Compared to other oxides, the author needs to explain or cite papers what advantages Ga2O3 has when used in gas sensor.

The aim of the given paper is not to compare Ga2O3 with other metal oxides with refer to gas sensing, moreover it has been already done in part by previous studies, which revealed high stability of gas sensor properties of this material. The authors of the present manuscript support this concept by adding information of the obtained materials gas sensor stability, which is represented on fig. 12 and corresponding considerations in the text of the manuscript:

“…Considering the long-term stability of obtained materials, however, it should be noted that they indeed show no degradation of response during continuous exploita-tion (Figure 12). Instead the opposite trend is observed – improvement of response af-ter many days of operation. This effect should be explained as the improvement of the gas sensing thick film layer consistency and intergrain contact number increase over the time of operation. As the XRD and TEM data has shown, the 300-500 oC temperature range is not enough to cause Ga2O3 grain growth, which may diminish the size effect and sensor response as a result. However, some minute surface diffusion processes are present and facilitated during repetitious surface partial reduction and further re-oxidation in the course of gas sensor response measurement towards reducing gases and VOCs. This process leads to formation of new “necks” between Ga2O3 grains, the electron transport through which is controlled by the presence of reducing gases and, thus, contributes to the sensor response [56-57]. Additional long term gas sensor measurements are needed to investigate whether these slow diffusion related processes may lead to grain and intergrain contact growth resulting in gas sensor response degradation.”

Reviewer 2 Report

In this study, authors introduced gas sensing properties of Nb(V) doped b-Ga2O3.  Although the study exploits the gas-sensitive properties of this material, this manuscript does not appear to be a study of gas sensors. The detail can be found as follow:

1. In the introduction, the author lacks an overview of gas sensors of Ga2O3 materials. What gas is this material most sensitive to?

2. The TEM image in figure 2 does not have a bar scale.

3. What do the results of determining the optical bandgap in Figure 5 help to explain the gas sensing properties of the investigated materials?

4. How does Nb(V) affect the gas sensing mechanism of Ga2O3?

5. The author did not compare the gas sensing data with other studies.

6. The conclusion is written in a very general way. The author needs to write down the parameters or results obtained from the study.

Author Response

Reviewer #2

  1. In the introduction, the author lacks an overview of gas sensors of Ga2O3 materials. What gas is this material most sensitive to?

The introduction has been rewritten and requested information has been added.

“…However, gallium oxide β-Ga2O3 shows n-type semiconductor properties only at high temperatures due to band gap width about 4.9 eV, while exact value varies depending on material physical state [15]. As a result, thin gas sensing films of Ga2O3 usually show their optimum performance towards either oxidizing or reducing gases, among which O2, NO, O3, CO, CH4 and other hydrocarbons, ammonia and volatile organic compounds only at high temperatures over 450 oC [16]. Efforts to decrease operating temperatures of gallia-based gas sensors are directed at composite sensitive materials formation [17-19], decoration with noble metals, possessing catalytic properties [20, 21], or exploitation of size effect of nanocrystalline high surface area Ga2O3-based materials [22-24]. The latter strategy may be considered as the most beneficial as the introduction of the second phase, either metal oxide or noble metal, in the materials structure can compromise the desired long-term stability of material as well as reliability of manufacturing process [25, 26]. Doping of Ga2O3 with n-type donor dopants – Sn, Ti, Si - has been reported to improve its electrical and gas sensing properties towards both oxidating and reducing gases due to activation of oxygen surface chemisorption [27-30]. Other n-type dopants have been reported in experimental and theoretical studies to drastically improve electrical properties of Ga2O3, valuable in semiconductor industry [31-33]. Among them Nb(V) has been proposed as a perspective and effective n-type dopant, being a shallow donor with small ionization energy and close ionic radii to Ga(III)in both octahedral and tetrahedral positions of monoclinic β-Ga2O3 lattice [34]. Further experimental studies have supported this concept [31, 35, 36].”

  1. The TEM image in figure 2 does not have a bar scale.

Actually there was a bar scale, however the authors respectfully admit, that it was quite small and not easy to read and refer to. The bigger and more convenient bar has been placed instead.

  1. What do the results of determining the optical bandgap in Figure 5 help to explain the gas sensing properties of the investigated materials?

Some additional considerations have been added to the text of manuscript:

“The band gap width decreases with the growth of Nb content due to formation of the NbGa substitution defects, which create either shallow donor levels beneath the conduction band or additional oxygen vacancies levels. The modern theoretical studies indicate, that the former case is more likely to be the cause of band gap narrowing [34]. Considering the gas sensing properties shallow donor levels, associated with the NbGa defects, should positively affect the free charge carrier concentration, which might be beneficial for oxygen chemisorption on the materials surface and as a result – for the gas sensor response itself [27-30].”

  1. How does Nb(V) affect the gas sensing mechanism of Ga2O3?

The gas sensor data and the discussion of it has been extended in the manuscript to show, that the improvement of gas sensor performance of Nb-doped materials compared to pure ones is of quantitative measure rather than qualitatiove, i.e. gas sensing mechanism is not affected by the doping in this study.

The Authors refer the Reviewer to new figures 9-12 and corresponding considerations.

  1. The author did not compare the gas sensing data with other studies.

the comparison table has been added, please see Table 3 and corresponding considerations

  1. The conclusion is written in a very general way. The author needs to write down the parameters or results obtained from the study.

the conclusion part has been extended:

“…The use of flame spray pyrolysis process allows to obtain β-Ga2O3 in ultrafine state with 6-12 nm grain size and the effective surface area more than 100 m2/g. Doping by Nb(V) during the synthetic procedure leads to both Nb incorporation in the lattice of β-Ga2O3 grains and formation of separate GaNbO4 phase if the temperature of the post-synthetic annealing goes above 800 oC. Doping of β-Ga2O3 by Nb(V) is accompa-nied by the charge compensation of the cationic sublattice by means of Ga(III) to Ga (I) transition. The combination of these phenomena does not allow to significantly im-prove electrical and gas sensor properties of ultrafine β-Ga2O3 by doping with elec-tron-donor cation Nb(V). The observed enhancement of the β-Ga2O3-based gas sensor performance can be connected with the increase of materials surface area due to the hampering of diffusion related grains growth and agglomeration at elevated working temperatures. The highest sensor response to low concentrations of reducing gases is observed in the case of Ga2O3 doped with 1% mol Nb(V), passed through 900 oC 24h postsynhetic annealing. The obtained ultrafine nanocrystalline samples are most suit-ed to detection of VOCs and allow to decrease working temperature and increase sen-sor response compared to other forms of Ga2O3 – thin films, nanowires or thick films. The manufactured sensing elements demonstrated improvement of gas sensor re-sponse during long term exploitation due to enhanced intergrain contacts and porous film integrity. Modification of the synthetic procedure is required in order to obtain Nb(V) doped Ga2O3 in ultrafine nanocrystalline state without Ga(III) to Ga (I) transi-tion.”

Reviewer 3 Report

The presented paper aims at investigating the structural, electrical and gas sensor properties of Nb(V) doped nanocrystalline β-Ga2O3. Despite an accurate description of the structural and electrical properties, the gas sensor properties are not accurately described and justified withe experimental data. I suggest improving this section as follows:

1) Please, explain why the resistance value of the material oscillates in Figure 7, while the temperature and the gas concentration should be constant. Which value could be selected to assess the gas sensor property of the material? 

2)  Data presented in Figure 8 could be a starting point to describe gas sensor properties, but it is not enough. Beside a change in resistance with the temperature, it must be shown at least a calibration curve for a selected gas, thus demonstrating that the resistance changes with the gas concentration.

3) In Figure 8, it is shown that this material could be employed to detect different gases. Please, demonstrate if this material could detect multiple gases in the same gas mixture and could measure their respective concentrations.

Author Response

Reviewer #3

  • Please, explain why the resistance value of the material oscillates in Figure 7, while the temperature and the gas concentration should be constant. Which value could be selected to assess the gas sensor property of the material?

The figure 7 has been redrawn and necessary explanations are added to the text.

  • Data presented in Figure 8 could be a starting point to describe gas sensor properties, but it is not enough. Beside a change in resistance with the temperature, it must be shown at least a calibration curve for a selected gas, thus demonstrating that the resistance changes with the gas concentration.

the requested data has been added through figures 9-11 and corresponding discussion has been added to the manuscript text

  • In Figure 8, it is shown that this material could be employed to detect different gases. Please, demonstrate if this material could detect multiple gases in the same gas mixture and could measure their respective concentrations

These suggestions of the respected Reviewer fall beyond the scope of the paper. The main objective of this study is to compare Nb-doped Ga2O3 with pure oxide in the ultrafine nanocrystalline state in terms of general sensor characteristics – value of sensor response, working temperature, baseline resistance, sensitivity to humid conditions, stability of response. Some considerations on the selectivity of gas sensor response are added to the conclusions section, however no particular practical task of certain gas detection in the presence of other compounds was an objective of the given study.

Reviewer 4 Report

This manuscript describes a Nb-doped Ga2O3 sensor prepared by a flame spray pyrolysis technique. The improvement of gas sensor properties was attributed to the decrease of band gap by either regulation donor levels or additional oxygen vacancies. This study showcases a potential method to improve the sensitivity of materials. However, there are still many shortcomings in performance research. The following comments need to be modified to meet the recruitment of the publication. 1. The gas sensor is used for gas detection, and the humidity problem is inevitable. The author should provide the gas sensitivity test data under different humidity conditions; 2. The author mentioned that Ga2O3 has good stability, but there is no relevant data to support it, please supplement the stability dataï¼› 3. In order to reflect the heterogeneity of the material properties, the authors should study the gas sensitivity of the material under different concentrations of gas and the detection limit of the sensorï¼› 4. Some references should be added and cited: Rare Metals volume 41, pages 1375–1379 (2022); Ceramics International Volume 41, Issue 10, Part B, December 2015, Pages 14790-14797

Author Response

Reviewer #4

  1. The gas sensor is used for gas detection, and the humidity problem is inevitable. The author should provide the gas sensitivity test data under different humidity conditions;

Figure 11 has been added to the manuscript in order to clarify this question.

  1. The author mentioned that Ga2O3 has good stability, but there is no relevant data to support it, please supplement the stability data

The requested data and considerations were added to the text (Fig.12 and corresponding text)

  1. In order to reflect the heterogeneity of the material properties, the authors should study the gas sensitivity of the material under different concentrations of gas and the detection limit of the sensorï¼›

The data on the gas sensor response dependence on the gas concentration is added to the manuscript.

  1. Some references should be added and cited: Rare Metals volume 41, pages 1375–1379 (2022); Ceramics International Volume 41, Issue 10, Part B, December 2015, Pages 14790-14797

With all respect to the esteemed Reviewer the authors could not find any ground to include the proposed articles in the reference list of the paper. The first one is related to thin solid films of Ga2O3 and their application in solar blind photodetectors, and is not related to the field of Ga2O3 doping or gas sensing at all. The second one deals with humidity sensing, which is effectuated through completely different electrical and chemical mechanisms compared to gas sensing and in this sense can barely contribute to the knowledge, discussed in the given manuscript.

Round 2

Reviewer 1 Report

The authors have revised the manuscript accordingly and the queries raised in the previous evaluation have also been well responded. The manuscript is now acceptable for publication.

Author Response

We are very grateful to the Reviewer for the valuable notes, which helped to improve the manuscript.

Reviewer 3 Report

I thank the authors for the big work done, but I still have some suggestions:

1) Since data are acquired at a fixed temperature, I would remove the second y-axis in 7a) and the yellow graph.

2) Please, since the trend could not be considered linear in Fig.10, please add more points in the calibration of the sensor. Adding more points, you could check for sensor linearity and try to provide a minimum detection limit. If the sensor response will not be linear, try to explain the reason.

3) Since multiple gas detection is beyond the scope of the paper, I suggest removing graphs related to the gas that you do not calibrate.

Author Response

  • Since data are acquired at a fixed temperature, I would remove the second y-axis in 7a) and the yellow graph.

The figure has been corrected in accordance with the reviewers requirements.

  • Please, since the trend could not be considered linear in Fig.10, please add more points in the calibration of the sensor. Adding more points, you could check for sensor linearity and try to provide a minimum detection limit. If the sensor response will not be linear, try to explain the reason.

The considerations on the linearity of sensor response alongside with the relevant literature references were added to the manuscript text. Considering the lower limit of response, the authors would like to gently notice, that the purpose of the present work is to investigate the effect of Nb doping on electrical, structural and gas sensor properties of nanocrystalline ultrafine gallia in the general sense, rather than to establish the sensor characteristics of obtained materials in detail in any practical case. That is why the manuscript is being prepared to publication in the “Materials” rather than in any specialized sensor journal. The authors would like to leave the detailed gas sensor properties characterization of the given Ga2O3 based ultrafine materials and those, which will be obtained in future, to more specified practical tasks in forthcoming articles in specialized chemical senor journals.

3) Since multiple gas detection is beyond the scope of the paper, I suggest removing graphs related to the gas that you do not calibrate.

The authors would like to leave the presented graphical data on the gas sensor performance as it is, since it is discussed in the corresponding part of the manuscript text in order to evaluate the electrical, structural and gas sensor effects of Nb doping on the ultrafine nanocrystalline Ga2O3.

Reviewer 4 Report

  • The answers to questions 1 and 2 are not satisfactory enough. First of all, the humidity study is too narrow, only up to 30%, so it is necessary to further test the influence of high humidity conditions on the gas-sensitive performance. Additionly, the stability curve tested by the authors is strange over time. Does this material have a future application?

Author Response

Reviewer #4-2

  • The answers to questions 1 and 2 are not satisfactory enough. First of all, the humidity study is too narrow, only up to 30%, so it is necessary to further test the influence of high humidity conditions on the gas-sensitive performance.

Additional measurements at 60% relative humidity were added to the plots and some additional considerations were introduced into the manuscript text.

  • Additionly, the stability curve tested by the authors is strange over time. Does this material have a future application?

Additional plot was added on the Figure 13 in order to simplify the perception and some brief considerations on the sensor applicability are introduced into the manuscript text.

Additional text is marked green.
